



# Differential effects of redox conditions on the decomposition of litter and soil organic matter

Yang Lin[1,2*], Ashley N. Campbell[3], Amrita Bhattacharyya[3,4], Nicole DiDonato[5], Allison M. Thompson[5], Malak M. Tfaily[5,6], Peter S. Nico[4], Whendee L. Silver[1*], and Jennifer Pett-Ridge[3*]

[1]Department of Environmental Science, Policy, and Management, University of California, Berkeley, California 94720
    [2]Department of Soil and Water Sciences, University of Florida, Gainesville, Florida 32611
    [3]Physical and Life Sciences Directorate, Lawrence Livermore National Laboratory, Livermore, California 94550
    [4]Earth Sciences Division, Lawrence Berkeley National Laboratory, Berkeley, California 94720
    [5]Environmental Molecular Sciences Laboratory, Pacific Northwest National Laboratory, Richland, Washington 99352
[6]Department of Environmental Science, University of Arizona, Tucson, Arizona 85719

*Correspondence to*: Yang Lin (ylin2@ufl.edu), Whendee L. Silver (wsilver@berkeley.edu), and Jennifer Pett-Ridge (pettridge2@llnl.gov)

## Abstract

Soil redox conditions exert substantial influence on biogeochemical processes in terrestrial ecosystems. Humid tropical forest
soils are often characterized by fluctuating redox dynamics, yet how these dynamics affect patterns in soil versus litter decomposition and associated $CO_2$ fluxes is not well understood. We used a [13]C-labeled litter addition to explicitly follow the decomposition of litter-derived vs. native soil-derived organic matter in response to four different soil redox regimes—static oxic or anoxic, and two oscillating treatments—in soil from the Luquillo Experimental Forest, Puerto Rico. We coupled this incubation experiment with high-resolution mass spectrometry to characterize the preferential decomposition of specific
classes of organic molecules. $CO_2$ production from litter and soil organic matter (SOM) showed distinctly different responses to redox manipulation. The cumulative production of SOM-derived $CO_2$ was positively correlated with the length of soil exposure to an oxic headspace ($r = 0.89$, $n = 20$), whereas cumulative [13]C-litter-derived $CO_2$ production was not linked to oxygen availability. The $CO_2$ production rate from litter was highest under static anoxic conditions in the first half of the incubation period, and later dropped to the lowest among all redox treatments. In the consistently anoxic soils, we observed
the depletion of more oxidized water-extractable organic matter (especially amino sugars, carbohydrates, and proteins) over time, suggesting that under anaerobic conditions, microbes preferentially used more oxidized litter-derived compounds during the early stages of decomposition. Results from kinetic modeling showed that more frequent anoxic exposure limited the decomposition of a slow-cycling C pool, but not a fast-cycling pool. Overall, our results demonstrate that substrate source— freshly added litter vs. native organic matter—plays an important role in the redox sensitivity of organic matter decomposition.
In soil environments that regularly experience redox fluctuations, anaerobic heterotrophs can be surprisingly effective in degrading fresh plant litter.



**Keywords**

Luquillo Experimental Forest, Puerto Rico, redox oscillation, greenhouse gas, soil respiration, soil organic matter, decomposition, FTICR-MS

## 1. Introduction

Tropical forests have some of the highest litter decomposition rates among terrestrial biomes (Parton et al., 2007) and are important sources of greenhouse gases, including carbon dioxide ($CO_2$), methane ($CH_4$), and nitrous oxide ($N_2O$) (Kirschke et al., 2013; Tian et al., 2016; Hashimoto et al., 2015). Climate change has the potential to alter the carbon (C) balance of tropical forests by altering their temperature and rainfall regimes (O'Connell et al., 2018). Yet our mechanistic understanding of how edaphic (temperature, moisture, redox, soil physical properties) and biotic (microbial community composition, functional potential) factors interact is lacking for these ecosystems with implications for understanding and predicting the effects of climate change.

Reduction-oxidation (redox) reactions are important drivers of decomposition and greenhouse gas emissions, particularly in warm, wet, fine-textured tropical soils. Oxygen ($O_2$) is a highly preferred terminal electron acceptor, and critical for the deconstruction of macromolecular organic compounds since the most powerful enzymes — oxidases and peroxidases — can only be used if $O_2$ is available (Sinsabaugh, 2010). Thermodynamic considerations predict that decreased $O_2$ availability should limit a soil's $CO_2$ production rate (Davidson et al., 2012; Greenwood, 1961). Thus, many modern biogeochemical models adopt a dimensionless scaling factor ($r_{oxygen}$) to account for decreased decomposition rates under anoxic conditions (Koven et al., 2013; Riley et al., 2014; Wania et al., 2009). Although upland soils are conventionally considered to be well aerated, recent studies have recognized the prevalence of hypoxic and anaerobic conditions in upland soils, at spatial scales ranging from aggregates and horizons to the entire soil profile (Hong et al., 2010; Fimmen et al., 2008; O'Connell et al., 2018; Hall et al., 2016a; Keiluweit et al., 2018; Silver et al., 1999; Schuur et al., 2001). Redox conditions can exhibit high variability across space and time in upland tropical forests (Hall et al., 2013; Silver et al., 1999; Liptzin and Silver, 2015) and impact multiple biogeochemical processes (Pett-Ridge and Firestone, 2005; Pett-Ridge et al., 2006; Chen et al., 2018a; Keiluweit et al., 2017). Intriguingly, some tropical soil lab and field studies have measured similar $CO_2$ emission rates during both oxic and anoxic periods (Teh et al., 2005; Liptzin et al., 2011; DeAngelis et al., 2010; Bhattacharyya et al., 2018), highlighting the need to better represent the redox sensitivity of decomposition dynamics in biogeochemical models.

When $O_2$ is limiting, soil microorganisms must use alternative terminal electron acceptors to oxidize organic matter via anaerobic metabolic pathways involving iron (Fe), sulfur (S) or manganese (Mn) reduction, denitrification, methanogenesis or fermentation. Iron reduction is recognized as a particularly important pathway of anaerobic respiration in tropical soils, given the abundance of Fe-minerals and Fe reducers in these environments (Ginn et al., 2017; Erickson et al., 2001; Hall et al., 2016a; Dubinsky et al., 2010). The activity of Fe reducers and other anaerobic microorganisms is strongly influenced by the chemical composition of their organic substrates. Thermodynamic theory predicts that the biogenic yield for the oxidation of a given





organic compound can be linked to its nominal oxidation state of C (NOSC) (Jin and Bethke, 2003; LaRowe and Van
Cappellen, 2011; Masiello et al., 2008). For example, reduced substrates such as alkenes and lipids (low NOSC values) should
not provide enough bioenergetic yield to be coupled with Fe reduction, whereas more highly oxidized substrates such as simple
organic acids (high NOSC values) can be readily oxidized by both Fe reducers and aerobic microorganisms (Keiluweit et al.,
2016). Therefore, substrate molecular composition likely regulates how the decomposition of organic matter responds to
dynamic redox conditions. More labile C inputs (e.g. root exudates, saccharides derived from plant tissues) may exhibit similar
rates of decomposition under oxic and anoxic conditions, because they can be readily utilized by both aerobic and anaerobic
microorganisms. In contrast, decomposition of more reduced substrates such as cutin and lipids may be limited by $O_2$
availability (Boye et al., 2017; Keiluweit et al., 2017), because anaerobic oxidation of these compounds is thermodynamically
unfavorable.

In this study, we used a laboratory incubation experiment to explore the effects of static and dynamic redox regimes on
plant litter vs. SOM decomposition and $CO_2$ production from a tropical forest soil. Two fluctuating redox regimes (high vs
low frequency) were imposed on the laboratory soil microcosms, in addition to static oxic and anoxic regimes. Labeled ([13]C)
grass litter was added to the soil incubations, allowing us to partition $CO_2$ fluxes into two sources, litter and soil organic matter
(SOM). Gas fluxes were analyzed with a data-model assimilation approach to compare redox effects on decomposition of a
fast-cycling vs. a slow-cycling C pool. We also characterized the chemical composition of water-extractable soil organic matter
using high-resolution mass spectrometry over the course of the experiment. We interpreted the disappearance of compounds
as a result of microbial decomposition/transformation and compared the preferential decomposition of specific compound
classes in response to different redox regimes. We predicted that: 1) $O_2$ availability would regulate the decomposition of both
litter and SOM in the same fashion, with $CO_2$ production positively correlated with the length of exposure to an oxic headspace,
and 2) decomposition of substrates characterized as 'more thermodynamically labile' would experience less severe $O_2$
limitation.

## 2. Materials and Methods

### 2.1. Experimental design

Surface soils (0-10 cm) (Humic Haploperox, Soil Survey Staff, 1995) were collected from near the El Verde field station
in the Luquillo Experimental Forest (LEF), Puerto Rico, part of the NSF-sponsored Long-term Ecological Research Program
and a NSF Critical Zone Observatory. The sampling location was positioned on a 15% slope where fluctuations in $O_2$
concentrations are common (O'Connell et al., 2018; Bhattacharyya et al., 2018); prior work in these soils has suggested their
microbial communities are adapted to redox oscillations that occur at least as often as every 4-6 days (Pett-Ridge and Firestone,
2005; Pett-Ridge et al., 2013; Pett-Ridge et al., 2006; DeAngelis et al., 2010). The study site is described locally as a tabonuco
(*Dacryodes excels* Vahl) forest, in the subtropical wet forest life zone according to the Holdridge life zone system (Ewel and





Whitmore, 1973), with a mean annual temperature of 23 °C, and mean annual precipitation of 3.5 m, relatively evenly distributed throughout the year (Scatena, 1989). The average soil pH, clay content, C concentration, and oxalate-extractable Fe concentration were 5.6, 20%, 6.1%, and 5.9 mg g$^{-1}$, respectively (O'Connell et al., 2018; Bhattacharyya et al., 2018).

Soils were shipped overnight to Lawrence Livermore National Laboratory, CA, and, upon arrival, were immediately homogenized by gently breaking down large aggregates and removing visible plant debris, rocks, and soil macro-fauna.
Approximately 20 g (oven dry weight equivalent, ODE) of soil was weighed into each of 44 glass microcosms (487 ml). Twenty of these were continuously monitored for $CO_2$ production and destructively harvested at the end of the experiment, while additional replicate microcosms were harvested during the experiment. All microcosms were first exposed to a 16-day ambient pre-incubation to allow soil respiration to stabilize. During the pre-incubation, all microcosms were flushed with humidified gas at approximately 3 ml/min following a redox regime that began with 4 days of oxic conditions (flushing the
headspace with compressed air), followed by 4 days of anoxic (flushing with $N_2$), and ended with another 4-day oxic/4-day anoxic cycle.

After the pre-incubation, all microcosms were amended with 180 mg $^{13}$C-labeled ground ryegrass litter (97 atom%, Isolife, Wageningen, Netherlands) and incubated for 44 days. This litter addition represented approximately 6% of the soil's native C content. Microcosms were split into four treatments: (1) static anoxic, (2) static oxic, (3) 4 days oxic/4 days anoxic (high
frequency; Fig. S1), (4) 8 days oxic/4 days anoxic (low frequency). For each, the headspace was manipulated in the same way as in the pre-incubation. Both high and low frequency treatments started and ended with oxic phases. In a companion study (Bhattacharyya et al., 2018), we used unlabeled litter with a similar experimental design to explore the coupled cycling of Fe and C and Fe geochemistry. We examined the effects of dynamic redox conditions on soil P dynamics in both labeled and unlabeled microcosms in another companion study (Lin et al., 2018).


## 2.2. Trace gas sampling and measurement

Headspace samples were collected approximately every 4 days during the incubation to assess fluxes of $CO_2$ and $^{13}$C-$CO_2$ concentrations. At each sampling point, a cohort of 20 microcosms (4 redox regimes × 5 replicates) was sampled immediately before the redox conditions were altered. To measure $CO_2$ fluxes, microcosms were temporarily sealed for 2 hours, and gas
samples were collected at the beginning and end of the sealed period by sampling 30 ml of the incubation jar headspace into pre-evacuated 20 ml vials. Microcosms were sealed for 3 hours when $CO_2$ production rates decreased towards the end of the experiment. Concentrations of $CO_2$ were measured at U.C. Berkeley on a gas chromatograph (GC-14A, Shimadzu, Columbia, MD), equipped with a thermal conductivity detector. Fluxes of $CO_2$ were determined by calculating the difference in $CO_2$ concentrations before and after the sealed period assuming a linear flux rate. An extra set of gas samples were collected after
the sealed period from each microcosm. In these samples, we measured the $^{13}$C/$^{12}$C isotope ratio of $CO_2$ with an isotope ratio mass spectrometer at U.C. Berkeley (IRMS; IsoPrime 100, Elementar, Hanau, Germany). We also measured $CH_4$ production





over the experiment, while the cumulative $CH_4$ production was trivial compared to $CO_2$ with no apparent effects of redox treatments (Fig. S2).

## 2.3 Microcosm harvests and chemical analysis

Soil microcosms were destructively harvested at three timepoints during the experiment. The subset of microcosms used for gas sampling was sampled on day 44 ($n = 5$ per treatment), and additional microcosms were sampled on days 20 and 36 ($n = 3$ per treatment). Microcosms from the low-frequency treatment were not harvested on day 36; instead, they were harvested on day 33, and not included in further analysis. Samples that had been exposed to an oxic headspace preceding the harvest were processed on the benchtop; those that had just finished an anaerobic period, along with those from the static anoxic

treatments, were processed in an anaerobic glove box (Coy Laboratory Products, Grass Lake, MI). Water-extractable dissolved organic C (DOC) was extracted by shaking 2 g of soil (dry weight equivalent) in 200 ml of Milli-Q water for 1 h and then passing through a 0.45 μm filter. Concentrations of DOC were measured using a Total Organic Carbon Analyzer (TOC-VCSH, Shimadzu). We also measured the $^{13}C/^{12}C$ isotope ratio of the background soil and litter using a Vario Micro elemental analyzer (Elementar, Hanau, Germany) in-line with the IRMS at U.C. Berkeley (Table S1).

For each harvested microcosm, a soil subsample was frozen and sent to the Environmental Molecular Sciences Laboratory at Pacific Northwest National Laboratory (Richland, WA) for Fourier-Transform Ion Cyclotron Resonance Mass Spectrometry (FTICR-MS) analysis of soil organic matter composition, following the approach described in Tfaily et al. (2017). In brief, 100 mg of soil (ODE) was sequentially extracted with water, methanol, and chloroform to extract polar and nonpolar extractable organic matter. Soil was shaken with each solvent (2 ml) for 2 hours on an Eppendorf Thermomixer and then

centrifuged before collecting the supernatant for further analysis. Water extracts were then desalted by solid phase extraction using Varian PPL cartridges according to Dittmar et al. (2008). After this clean up step, samples were eluted in methanol and injected directly onto the FTICR-MS. The methanol extract fraction was injected directly onto the mass spectrometer whereas the chloroform fraction was mixed with methanol (1:1 ratio) before injecting onto the mass spectrometer. A 12 T Bruker SolariX FTICR spectrometer was used to collect high resolution mass spectra of the organic compounds in each extract.

Electrospray needle voltage and Q1 were set to +4.4 kV and $m/z$ 50, respectively, with 144 scans collected and averaged for each sample. Mass spectral peaks were internally calibrated using a commonly identified OM homologous series in which each member of the series differs from the next/previous by one –$CH_2$ group or 14 mass units (Da)(e.g., separated by 14 Da). The mass measurement accuracy was within 1 ppm for singly charged ions across a broad m/z range ($m/z$ 200-800). Peaks were extracted with the DataAnalysis software (Bruker, Billerica, MA) using the following parameters: signal to noise

threshold >= 7, relative threshold base peak = 0.01, absolute intensity threshold = 100, and the m/z range of 200-750. Molecular formulas were  assigned using EMSL's in-house software, Formularity, based on the Compound Identification Algorithm (CIA) (Kujawinski and Behn, 2006; Tolić et al., 2017). Spectra data were processed with the R package 'ftmsRanalysis' (Bramer and White, 2019). In two out of the three microcosms from the day 20 high-frequency treatment, the FTICR-MS





returned an extremely low number of peaks (< 200); this specific treatment on day 20 was excluded from further analyses.

Two additional samples from the high-frequency treatment were also excluded due to low number of peaks: one on day 36 and one on day 44. To identify potential outliers, we computed robust Mahalanobis distance (rMd) from peak abundance data using the 'rmd_filter' function in the R package 'pmartR' (Stratton et al., 2019). The rMd data were then mapped to $P$ values using five metrics, including correlation coefficient, fraction of missing data, median absolute deviation, skewness, and kurtosis (Matzke et al., 2011). Using α = 0.01, we found one potential outlier from the static anoxic treatment on day 44.

Further analyses revealed that this sample obviously deviated from all others in terms of organic matter composition (Fig. S3). Thus, we excluded it from downstream FTICR-MS analysis. Our analyses showed that redox treatment did not noticeably affect the composition of methanol- or chloroform-extractable organic matter. Thus, we focused on the results of water-extractable organic matter, which were most affected by redox treatments and are expected to be more relevant to the microbial processes naturally occurring in this wet tropical soil.

**2.4. Modeling total CO₂ fluxes**

To evaluate the effects of redox treatments on decomposition rates, we fitted the total $CO_2$ fluxes (without isotopic partitioning) from each redox treatment and timepoint to a two-pool model with three coefficients (Sierra and Markus, 2015). This model assumes that soil C is comprised of two distinct pools with different decomposition rates following first-order kinetics:

$$\frac{dC}{dt} = -k_1 * C_1 - k_2 * C_2$$

$$C_1 = \gamma * C_0$$

where the total amount of C in the microcosm was made up of "fast-cycling" and "slow-cycling" pools ($C_1$ and $C_2$, respectively), with corresponding decomposition constants $k_1$ and $k_2$ ($k_1 > k_2$), respectively and expressed as day$^{-1}$. We use the convention of 'fast-cycling' and 'slow-cycling' while acknowledging that organic matter is more realistically represented by

a continuum of material with different cycling rates. The initial C content ($C_0$) was partitioned into $C_1$ and $C_2$ using the coefficient $\gamma$. Since we did not use isotopic data in this part of the analysis, the model-derived partitioning between the fast-cycling and slow-cycling pools is independent of that between litter- and SOM-derived organic matter. The fast-cycling pool primarily contributes to the initial peak of the $CO_2$ flux during the incubation, while the slow-cycling pool accounts for the $CO_2$ flux when it stabilizes after the peak. Because the values of all three model coefficients varied amongst redox treatments,

we were able to independently compare redox effects on the size of the fast-cycling and slow-cycling pools and their degradability. Model coefficients were estimated using a two-step procedure described in Soetaert and Petzoldt (2010). First, an initial approximation to the coefficients values was estimated by optimizing a cost function based on the sum of squared residuals of coefficients weighed by the number of observations. Functions 'modCost' and 'modFit' from the R package of 'FME' were used to construct the cost function and to derive the coefficients estimates. Second, the estimated coefficients and

the covariance matrix were used as priors for a Bayesian optimization procedure using Markov chain Monte Carlo with the



function 'modMCMC'. A total of 20000 iterations were run with a burn-in of 1000 iterations. We report the mean estimates and standard deviations of each coefficient obtained from the Bayesian procedure (Sierra et al., 2017).

## 2.5. Data analysis

The percent contribution of $^{13}$C litter ($P_{litter}$) to the total $CO_2$ flux was calculated using a linear mixing model:

$$P_{litter} = 100 * \frac{\chi(^{13}C)_{flux} - \chi(^{13}C)_{SOM}}{\chi(^{13}C)_{litter} - \chi(^{13}C)_{SOM}}$$

Here, $\chi(^{13}C)_{flux}$ represents the measured atom% of $^{13}$C of the $CO_2$ flux. Variables $\chi(^{13}C)_{litter}$ and $\chi(^{13}C)_{SOM}$ refer to the measured atom% of the labeled litter and SOM (97 and 1.08 atom%, respectively; Table S1). This calculation was conducted on each microcosm, thus allowing us to estimate the errors of $P_{litter}$. Flux rates were assumed to change linearly between two sampling days, and cumulative gas fluxes were calculated as integrals over the 44-day period (area under the curve). The effect of the treatments on cumulative gas fluxes was examined by analysis of variance (ANOVA) followed by Tukey's tests using the R package 'emmeans' (Lenth, 2019). Treatment effects on the $CO_2$ fluxes were also studied following this approach on each sampling day.

We used regression models to evaluate the effects of headspace redox conditions (oxic vs. anoxic headspace) on $CO_2$ fluxes in each of the two fluctuating redox treatments. As $CO_2$ production derived from litter had a strong decreasing trend over time, its regression model included three terms: the decreasing trend (modeled by a quadratic polynomial function of time (Diggle et al., 2002)), the effects of headspace redox conditions, and an error term. Preliminary analysis suggested that the decreasing trend alone explained of 66% and 73% of the variability (i.e., $R^2$ of the regression) in the high-frequency and low-frequency treatment, respectively. We tried to account for the autocorrelation between measurements made from the same jar microcosm by designating its effect as a random intercept term; however, this did not improve the model fit. Thus, microcosm was not included as a variable in the regression model. As $CO_2$ production from SOM did not show a clear temporal trend, we treated time as a random intercept in the model along with the effects of redox and error. Again, we did not include microcosm effects because model fit was not improved.

Nonmetric Multidimensional Scaling (NMDS) was used to visualize the effects of redox treatments on organic matter composition. Abundance data derived from FTICR-MS peak intensities were first converted to presence/absence data. Then NMDS analysis was conducted using the R 'vegan' package to search for a minimal stress solution based on Jaccard distance (Oksanen et al., 2019). We also used presence/absence data to compare the relative abundances of compound classes between redox treatments. This approach assumes equal concentrations for all compounds with assigned peaks and overcomes the potential bias associated with different ionization efficiency between compound classes (Boye et al., 2017). We compared the treatment effects using a permutational multivariate ANOVA (PERMANOVA) with the 'adonis' function in the 'vegan' package.

Using the stoichiometry of the assigned formula, the NOSC value was calculated for each mass peak in the FTICR mass spectra following the equation (LaRowe and Van Cappellen, 2011):



$$NOSC = \frac{-Z + 4C + H - 3N - 2O + 5P - 2S}{C} + 4$$

where $Z$ corresponds to the net charge of the compound (which is assumed to be zero), and $C$, $H$, $N$, $O$, $P$, and $S$ represent the

stoichiometric numbers of each respective element. Molecular formulas were also assigned to major biochemical compound classes (i.e., amino sugar, carbohydrate, condensed hydrocarbon, lignin, lipid, protein, tannin, unsaturated hydrocarbon, and other) following Kim et al. (2003).

We compared the chemical composition of water-extractable organic matter between days 20 and 44 of the experiment by grouping all the detected compounds into three categories: observed on both days, uniquely found on day 20, and uniquely

found on day 44. This classification was achieved using a G-test (Webb-Robertson et al., 2010) with $P$ value of 0.05 via the 'uniqueness_gtest' function in the 'ftmsRanalysis' package (Bramer and White, 2019). In order to be considered as present, one formula must be present in at least 2 replicates on a given day (Bramer and White, 2019). This criterion is necessary as it helped to remove rare peaks that could greatly skew the results, while it did reduce the number of compounds included in this analysis compared to the total number of identified compounds (Table S2). We assumed that the compounds unique to day 20

(i.e. not detected in day 44 samples) had been degraded, modified by microbes, or became associated with minerals towards the end of the experiment, while the compounds unique to day 44 (i.e. not detected in day 20 samples) were newly produced by microbes, released from macromolecules, or desorbed from minerals in this time frame (Chen et al., 2018b).

## 3. Results

### 3.1. Gas fluxes

We used $^{13}C\text{-}CO_2$ abundance to partition total $CO_2$ production between litter- and SOM-derived pools and found distinct responses from these two $CO_2$ sources to our 44-day redox manipulation (Fig. 1a,b). The length of exposure to an oxic headspace (measured in days) was positively correlated with the cumulative $CO_2$ produced from SOM ($r = 0.89$, $P < 0.001$, $n = 20$), but not the $CO_2$ produced from litter ($r = -0.35$, $P = 0.14$, $n = 20$). Cumulative $CO_2$ fluxes derived from SOM were lowest in the static anoxic treatment and highest in the static oxic treatment (Tukey's tests: $P$s $< 0.05$); $CO_2$ respired from the

static anoxic soils was only $65 \pm 3\%$ of the cumulative flux measured in the static oxic conditions (Tukey's test: $P < 0.001$). SOM-derived cumulative $CO_2$ was intermediate in the fluctuating treatments, significantly higher than the static anoxic treatment, and lower than the static oxic treatment (Tukey's tests: $P$s $< 0.05$).

In contrast, the cumulative $CO_2$ production from added plant litter was highest in the static anoxic treatment, and significantly greater than in the two fluctuating treatments by at least $19 \pm 6\%$ (Tukey's tests: $P$s $< 0.05$, Fig. 1a). The static

oxic treatment had $20 \pm 7\%$ higher litter $CO_2$ production relative to the high frequency treatment (Tukey's test: $P = 0.05$) but was not statistically different from the low frequency treatment or static anoxic treatment (Fig. 1a). Averaging across all treatments, approximately 35% of the added litter C was lost via $CO_2$. The combined litter- and SOM-derived $CO_2$ (i.e., total





$CO_2$) production was highest in the static oxic treatment (Tukey's test: $P < 0.001$), and not statistically different amongst the static anoxic, high frequency, and low frequency treatments (Fig. 1c).

The litter-derived $CO_2$ fluxes showed distinct responses to redox treatments over time: during the early part of the experiment, they were significantly higher in the static anoxic treatment relative to the other redox treatments (Fig. 2; $Ps < 0.05$ for days 5 and 8; $Ps <= 0.19$ for days 12 and 16). After three weeks of incubation, litter $CO_2$ production had markedly declined in all four treatments, and by the final two sampling timepoints (days 41 and 44), litter $CO_2$ production was significantly lower in the static anoxic treatment than in other treatments (Tukey's tests: $Ps < 0.05$, Fig. 2).

Unlike litter decomposition, the decomposition of SOM did not show clear temporal trends during the 44-day incubation, except in the static anoxic treatment, where the rates significantly decreased with time ($r = -0.66$, $P < 0.001$, $n = 60$) (Fig. 2). In the latter half of the experiment (after the first 3 weeks), SOM-derived $CO_2$ fluxes from the static anoxic soils declined and became significantly lower than those measured in the static oxic soils (Tukey's tests: $Ps < 0.05$). Compared to the two fluctuating redox treatments, the static anoxic soils also had lower SOM-derived $CO_2$ production on days 28, 36, 41, and 44

(Tukey's tests: $Ps < 0.05$).

Both litter- and SOM-derived $CO_2$ fluxes responded significantly to the 4-day and 8-day shifts in redox conditions imposed during our two fluctuating treatments (Fig. 3). Average $CO_2$ production from SOM was $29 \pm 12\%$ and $32 \pm 15\%$ lower when measured at the end of an anoxic period vs. the end of an oxic period in the high-frequency and low-frequency treatments, respectively ($Ps < 0.05$). After accounting for the general decreasing trend with time (that explained over 65% of the

variability), average litter $CO_2$ production was $36 \pm 12\%$ lower when measured following an anoxic period vs. an oxic one in the high-frequency treatment ($P < 0.05$). A similar trend was also found in the low-frequency treatment, although it was not statistically significant ($P = 0.13$).

### 3.2. Modeling total $CO_2$ fluxes

We fit the total $CO_2$ flux (litter + SOM) to a two-pool model that characterizes decomposition using first-order kinetics of

two soil C pools with distinct decomposition rates, and found that redox treatments affected both the pool sizes and their decomposition rates (Fig. 4). The size of the fast-cycling pool (determined by the coefficient $\gamma$), was the primary contributor to the initial $CO_2$ flux during the experiment and was substantially larger in the static anoxic treatment relative to the other treatments. However, decomposition rates of this fast pool ($k_1$, thought to cycle on the order of days) were highly variable and thus showed no significant response to anoxic conditions. In our four soil redox treatments, increased exposure to oxygen

corresponded with an increase in the decomposition rate of the slow-cycling pool ($k_2$), which is thought to turn over on the orders of years.





### 3.3 Chemistry of water-extractable organic matter

Redox oscillation patterns significantly influenced the chemical composition of water-extractable organic matter according to NMDS and PERMANOVA analyses ($P < 0.05$; Fig. 5). While water-extractable organic matter in the static anoxic treatment appeared to be chemically distinct from those from other treatments (especially midway through the incubation, on day 20), we did not observe a clear separation between the static oxic and two fluctuating treatments. Redox effects on compound chemistry were evident by examining their NOSC values (Fig. 6). On day 20, the mean NOSC value of water-extractable organic matter was higher in the static anoxic treatment than in the static oxic treatment (Tukey's test: $P < 0.05$) and tended to be higher than in the low-frequency treatment (Tukey's test: $P = 0.12$); however, no difference in mean NOSC values were detected among redox treatments on either day 36 or 44. Mean NOSC value did not change over time between day 20 and 44.

When we analyzed the proportion of compounds that were unique to each harvest point, a large fraction of unique water-extractable compounds were present on day 20 in the static anoxic treatment (Fig. 7): 320 out of 1052 compounds were unique to day 20 in the static anoxic treatment, suggesting these compounds disappeared between days 20 and 44, while a small number of compounds ($n = 32$) were only found on day 44. However, in the static oxic treatment, most of the compounds that were present at day 20 were also extracted at day 44 (703 out of 785). Only 1.5% of compounds ($n = 12$) were unique to day 20 in the static oxic treatment. Similarly, in the low-frequency treatment, most of the compounds (704 out of 837) were observed on both days 20 and 44, with a small fraction ($n = 94$) unique to day 20. Combining data from days 20 and 44, the total number of unique compounds detected in the classes of amino sugar-like, carbohydrate-like, protein-like, and other was higher in the static anoxic than in the static oxic treatment. In these four compound classes, over half of the compounds were uniquely identified on day 20 in the static anoxic treatment. We did not conduct this analysis for the high-frequency treatment due to the low data quality of these samples on day 20.

Redox effects on compound chemistry were also apparent when we examined the relative abundances of compound classes (Fig. S4); on day 20, carbohydrate- and protein-like compounds were enriched in the static anoxic treatment compared to other treatments (Tukey's tests: $Ps < 0.05$); the relative abundance of amino sugar-like compounds in the static anoxic treatment was also higher than that in the static oxic treatment (Tukey's test: $P < 0.05$) and marginally higher than that in the low-frequency treatment (Tukey's test: $P < 0.10$). The relative abundance of lignin-like compounds was also slightly lower in the static anoxic treatment than in other treatments (Tukey's tests: $Ps < 0.10$). In contrast, redox treatments did not affect the abundance of any compound class on days 36 (data not shown) or 44. We note that these redox effects on compound chemistry were not driven by changes in total DOC concentrations (Fig. S5), as DOC concentrations were higher in the static anoxic treatment than in other treatments on both day 20 (Tukey's tests: $Ps < 0.05$) and day 44 (Tukey's tests: $Ps < 0.01$).





## 4. Discussion

The isotopic partitioning allowed by our tracer study revealed distinct effects of redox manipulations on the decomposition of plant litter and SOM, in contrast to our first prediction that low $O_2$ concentrations would decrease both litter and SOM decomposition. Surprisingly, litter decomposition rates were the highest under static anoxic conditions during the early part of
the experiment, suggesting that even soil anaerobes were able to effectively utilize some of the litter-derived compounds. In the static oxic and fluctuating redox soils, litter $CO_2$ fluxes were high immediately after the litter addition, but then gradually declined over time. In contrast, litter $CO_2$ fluxes in the static anoxic soils did not peak until 5 days after the litter was added, and remained at a higher level than other treatments until day 20. These patterns suggest that the static anoxic conditions permitted the degradation of labile, litter-derived compounds to occur over a longer period of time. The easily degradable
compounds derived from litter were mostly depleted by day 20, and nearly exhausted by the end of the 44-day experiment, when the litter decomposition rates in all treatments were very low (and lowest in the static anoxic soils). This coincided with the disappearance of over half of the amino sugar-, carbohydrate-, and protein-like compounds between days 20 and 44 under static anoxic conditions.

Together, the gas fluxes and chemistry data indicate that these amino sugar-, carbohydrate-, and protein-like compounds
were most likely derived from litter and were preferentially degraded by anaerobic decomposers in the anoxic treatment. Decomposition of labile compounds can be readily coupled with fermentation and substrate-level phosphorylation, while the lack of $O_2$ inhibits the synthesis of peroxidases and oxidases, and therefore limits the depolymerization of complex compounds such as lignin and tannin (Sinsabaugh, 2010; Megonigal et al., 2004; Reineke, 2001). Chen et al. (2018b) also reported that sugar-, carbohydrate-, and protein-like compounds were preferentially utilized by microbial decomposers during an anoxic
incubation of arctic soils.

Our FTICR-MS analysis further shows that the chemical composition of compounds (e.g., their oxidation state) influences their degradability under distinct redox conditions. Halfway through our study, the mean NOSC value of water-extractable organic matter was higher in the static anoxic soils than in other treatments, and declined in subsequent days. This change in mean NOSC value coincided with the disappearance of sugar-, carbohydrate-, and protein-like compounds between days 20
and 44 under static anoxic conditions, suggesting that anaerobes preferentially decomposed litter-derived compounds with relatively high NOSC value during the first half of our incubation. This result is consistent with the widely-accepted prediction that reduced compounds with low NOSC values are not thermodynamically favorable to be coupled with anaerobic metabolism (Jin and Bethke, 2003; LaRowe and Van Cappellen, 2011; Boye et al., 2017). Even though the rapid litter decomposition under anoxic conditions was specific to the early stages of decomposition, we expect that similar phenomena may occur under field
conditions, because *in situ* soils constantly receive fresh C inputs from litter and root exudates.

In contrast to litter decomposition, native SOM decomposition was limited by the exposure to anoxic conditions. The SOM likely contained a lower proportion of easily degradable compounds that would be preferentially targeted by microbial decomposers at the beginning of the decomposition process. Our results support the interpretation that the chemical



composition of organic matter influences how decomposition processes are impacted by redox conditions. In this humid
tropical forest soil, aerobic and anaerobic heterotrophs both appeared to be effective in degrading the labile substrates enriched
in fresh plant litter, while aerobic decomposers were more effective in breaking down the relatively refractory compounds
found in SOM, as expected based on the role of $O_2$ in oxidative activity. Similar effects of substrate liability have been reported
in marine sediments and wetlands (Hulthe et al., 1998; Kristensen and Holmer, 2001; De-Campos et al., 2012), but rarely in
upland soils that often experience oscillation in redox conditions. Interestingly, SOM decomposition rates under anoxic
conditions were approximately 65% of that under oxic conditions (i.e., $r_{oxygen}$ = 65%), which is higher than the $r_{oxygen}$ values
prescribed in numerical models (2% - 40%; reviewed by Keiluweit et al., 2016). This result indicates that the microbial
community in this soil may be well adapted to anaerobic conditions (Pett-Ridge and Firestone, 2005; Pett-Ridge et al., 2013;
DeAngelis et al., 2010), likely driven by the dynamic redox conditions that characterize these soils (Liptzin and Silver, 2015;
Silver et al., 1999).

Results from our modeling analysis also suggest that the decomposition of refractory C was more limited by $O_2$ availability
than that of labile C, as the decomposition rate of the slow-cycling pool declined under an anoxic headspace, while that of the
fast-cycling pool did not. As noted earlier, the partitioning of total C in our microcosms between the fast-cycling and slow-
cycling pools was independent of that between litter and SOM. Thus, our modeling and isotopic evidence independently
support that substrate source and composition influence the redox sensitivity of organic matter (Boye et al., 2017; Keiluweit
et al., 2017). Our modeling analysis also shows that the size of the fast-cycling pool was larger in the static anoxic treatment
than in other treatments, which is consistent with the observation that a high litter decomposition rate was maintained for an
extended period time under anoxic conditions. Together our results highlight the importance for biogeochemical models to
prescribe different $r_{oxygen}$ to soil C pools based on their chemical composition and degradability.

    Periodic anoxic events consistently decreased litter and SOM decomposition rates in our two fluctuating treatments,
suggesting that even brief exposure to anoxia inhibited organic matter decomposition. These patterns contrast with our finding
that static anoxic conditions increased litter decomposition relative to static oxic conditions early in our incubation. These
results demonstrate that the duration and frequency of low redox events regulate their impacts on decomposition processes.
We suspect that redox fluctuation would not be favorable for strictly aerobic or anaerobic heterotrophs, meaning that neither
group was able to achieve their full potential in organic matter degradation. Alternatively, the periodic anoxic events were too
short for strictly anaerobic heterotrophs to establish their community. Redox oscillation might also promote facultative
anaerobic microbial communities. These microbes might have lower growth potential compared to strictly aerobic/anaerobic
taxa, as predicted by the recently discovered trade-off between stress tolerance and growth performance (Maynard et al., 2019).
Recent studies also found that Fe(II) oxidation, which is typically rapid in fluctuating redox environments, could preferentially
protect the aromatic lignin compounds of plant litter from decomposition (Hall et al., 2016b; Wang et al., 2017). The relative
importance of the microbial and geochemical mechanisms listed above is a fruitful venue for future research, as the importance
of frequent redox oscillation is increasingly being recognized in upland ecosystems (Ginn et al., 2017; Keiluweit et al., 2016;
Barcellos et al., 2018; Lin et al., 2018).



Climate models predict increasing frequency and intensity of extreme weather events in the tropics, ranging from droughts to large storms and hurricanes, as a result of climate change (Patricola and Wehner, 2018; Chadwick et al., 2015). These

extreme events would trigger significant changes in soil redox conditions (Wieder et al., 2011; O'Connell et al., 2018), although the effects on belowground biogeochemical processes are poorly quantified. For example, extended drought can enable prolonged soil aeration, while large storm or hurricane events can greatly increase litter inputs to soil and sustain anoxia by increasing soil water content and microbial respiration. Our results suggest that the decomposition of plant-derived, labile organic matter, such as amino sugar and carbohydrate, would largely be unaffected by the changes in redox conditions

associated with extreme events. On the other hand, the decomposition of more refractory organic compounds in litter and SOM, such as those with low NOSC values, would likely be limited during periods of high rainfall and low redox events. Drought periods could favor the decomposition of these refractory compounds by aerobic decomposers as long as water availability would not be too low to inhibit microbial activity. Our results help to reduce the uncertainties surrounding how decomposition processes may respond to the extreme events and climate change.

In conclusion, we found distinct responses of litter and SOM decomposition to different redox conditions in a humid tropical forest soil. Litter decomposition rates under static anoxic conditions were surprisingly high during the first two weeks following litter additions, which was likely maintained by the decomposition of compounds with high NOSC values, particularly amino sugars, carbohydrates, and proteins. In contrast, oxygen availability limited the extent of SOM decomposition under low and fluctuating redox conditions, which was consistent with the results of our kinetic modeling.

Overall, static anoxic conditions maintained 65% of the SOM decomposition rate measured under static oxic conditions, indicating that soil anaerobic heterotrophs in this humid tropical soil are surprisingly effective at degrading native SOM. Together our results indicate that substrate composition and source influences how redox conditions affect belowground decomposition processes and are important variables to consider in numerical models of greenhouse gas fluxes from tropical forests.


**Code/Data availability**

Data from this study will be made available via the Luquillo CZO and Hydroshare (http://www.hydroshare.org/) upon acceptance for publication.

**Author contributions**

A.N.C. and J.P.R. designed the incubation experiment; Y.L. performed the gas analysis with help from A.N.C. and A.B; Y.L. conducted modeling analysis; M.M.T. performed the FTICR-MS analysis; Y.L. analyzed the FTICR-MS data with help from N.D., A.M.T., and M.M.T; J.P.R., W.L.S. and P.S.N. provided intellectual expertise; Y.L. led the manuscript development with contribution from all other coauthors; J.P.R. and W.L.S. provided funding for experimental set-up and salary

support.



## Acknowledgements

We thank Daniel Nilson, Elizabeth Green, Jessica Wollard, Shalini Mabery, Rachel Neurath, Keith Morrison, Christopher Ward, Jeffery Kimbrel, Steve Blazewicz, Erin Nuccio, Mona Hwang, Feliza Bourguet, Summer Ahmed, Heather Dang, Kana Yamamoto, and Sally Hall for assistance in the laboratory and in the field. We thank Nikola Tolic, Rosey Chu, and Jason
Toyoda for their help with FTICR-MS analysis. Avner Gross, Ljiljana Pasa-Tolic, and Allegra Mayer provided advice and/or helpful conversations. This project was supported by a US Department of Energy Early Career Research Program Award to J. Pett-Ridge (SCW1478) administered by the Office of Biological and Environmental Research, Genomic Sciences Program, and EMSL awards 48643, 48477, 48650, 48832 to J. Pett-Ridge. Work at LLNL was performed under the auspices of the U.S. Department of Energy under Contract DE-AC52-07NA27344. Work at UC Berkeley was supported by DEB-1457805 (WLS),
Luquillo CZO (EAR-1331841), and LTER (DEB-0620910). WLS was also supported by the USDA National Institute of Food and Agriculture, McIntire Stennis project CA-B-ECO-7673-MS. A portion of the research was performed using EMSL (grid.436923.9), a DOE Office of Science User Facility sponsored by the Office of Biological and Environmental Research.

## Competing interests

The authors declare that they have no conflict of interest.

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









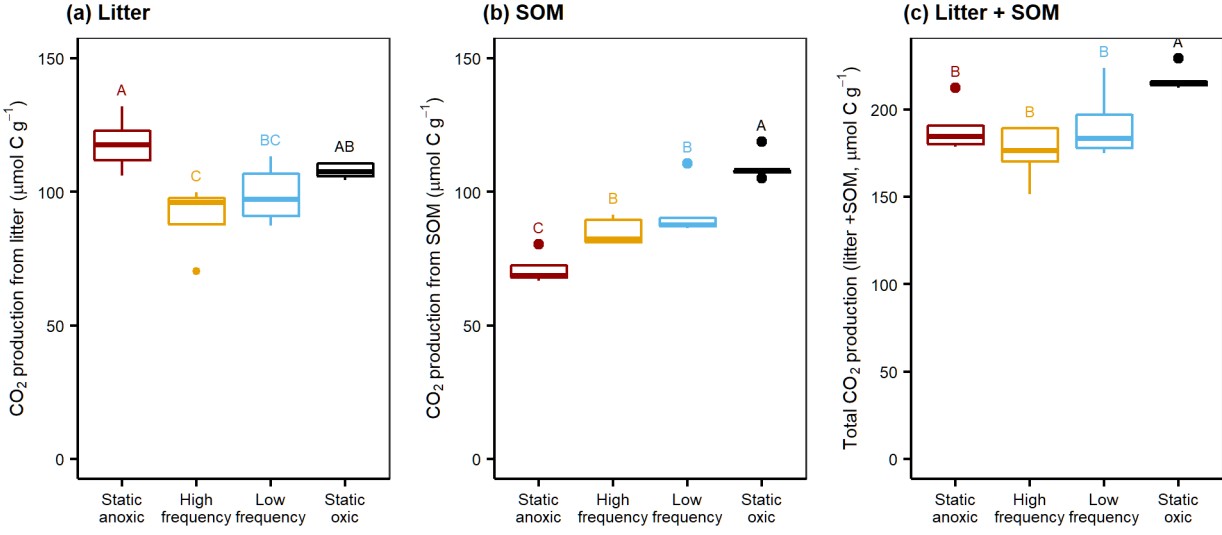


**Fig. 1. Effects of four redox treatments on the cumulative $CO_2$ production from litter (a), soil organic matter (SOM; b), and litter and SOM (c) for a 44 day redox incubation of tropical ultisols from the Luquillo Experimental Forest, Puerto Rico. $CO_2$ flux was partitioned between litter and SOM using $^{13}C$-$CO_2$ abundance and an isotopic mixing model. Boxplot whiskers represent 1.5 times the interquartile range of data. Different letters indicate significant difference at $\alpha = 0.05$ level (Tukey's tests). From left to right,**
**redox treatments had increased exposure to headspace oxygen. $n = 5$ per treatment. Static anoxic and oxic conditions were maintain by flushing incubation headspace with $N_2$ or air, respectively; "high" and 'low' frequency fluctuation incubations oscillated between redox states (4 days oxic/4 days anoxic (high frequency), oxic 8 days oxic/4 days anoxic (low frequency)). Note that the scale of $CO_2$ production is different in panel (c).**






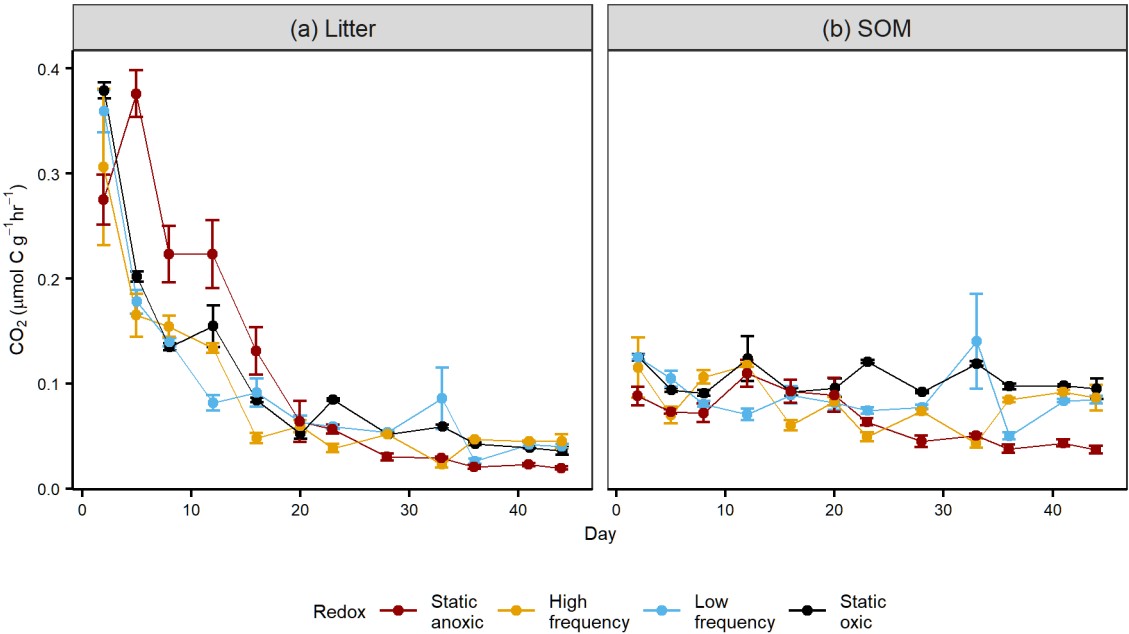

**Fig. 2. Effects of four redox treatments on the instantaneous production rate of $CO_2$ derived from $^{13}C$-litter (a) and soil organic matter (SOM, b) in a 44 day incubation study of wet tropical soils from the Luquillo Experimental Forest, Puerto Rico. Error bars indicate standard errors of the means. $n = 5$ per timepoint and treatment.**




Biogeosciences Open Access
Discussions
EGU



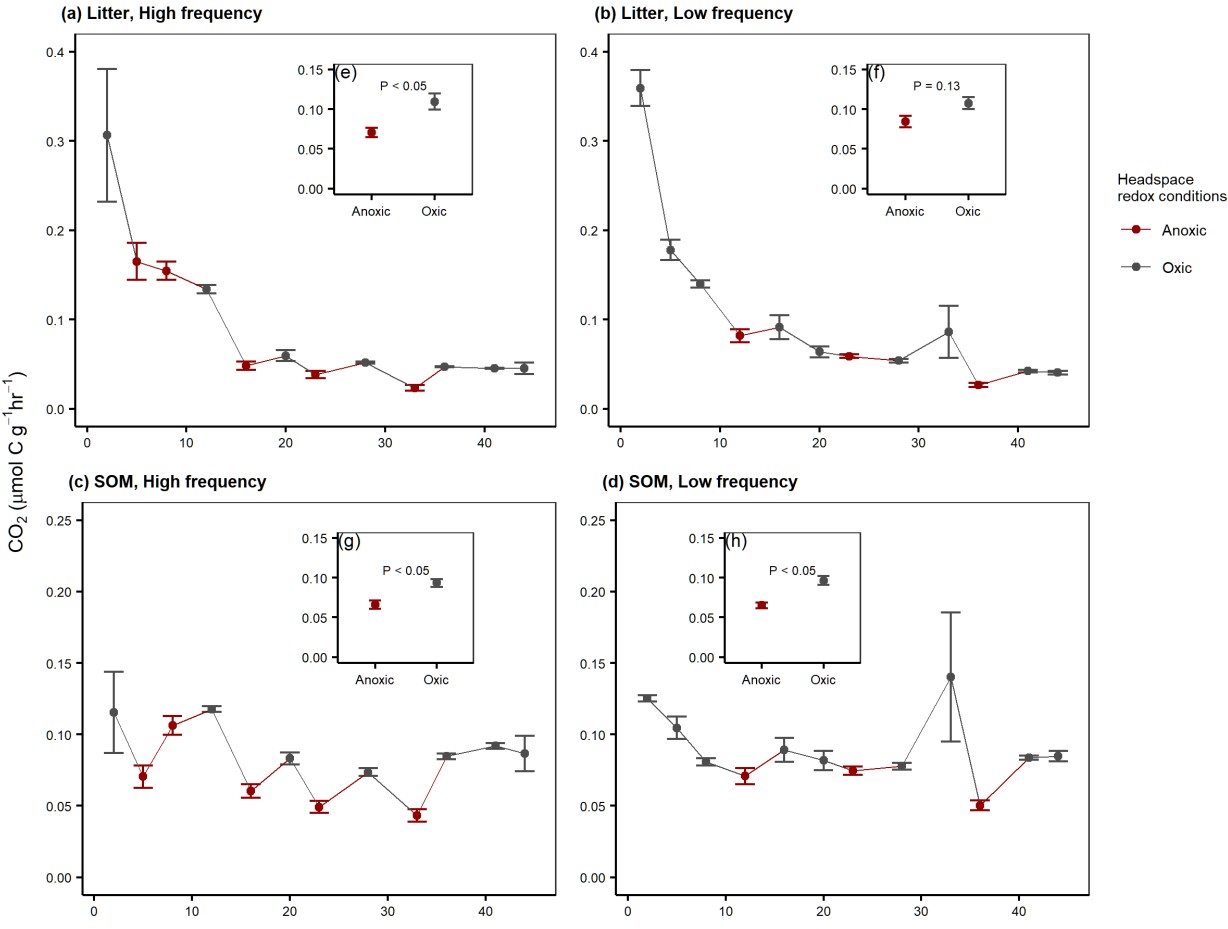

**Fig. 3.** Effects of headspace redox conditions (anoxic vs. oxic) on $CO_2$ production from litter (a and b) and soil organic matter (SOM, c and d) in high-frequency (a and c) and low-frequency (b and d) redox oscillation treatments during a 44 day incubation study. Error bars indicate standard errors of means. $n = 5$ per timepoint and treatment. Insets compare the means and standard errors of CO₂ production under anoxic vs. oxic headspace with $P$ values. Litter $CO_2$ production in the inset figures have been adjusted against their decreasing trend over time, which was modeled as a quadratic polynomial function. See Methods for details of statistical analyses. Note that the scale of $CO_2$ production was different between litter and SOM.





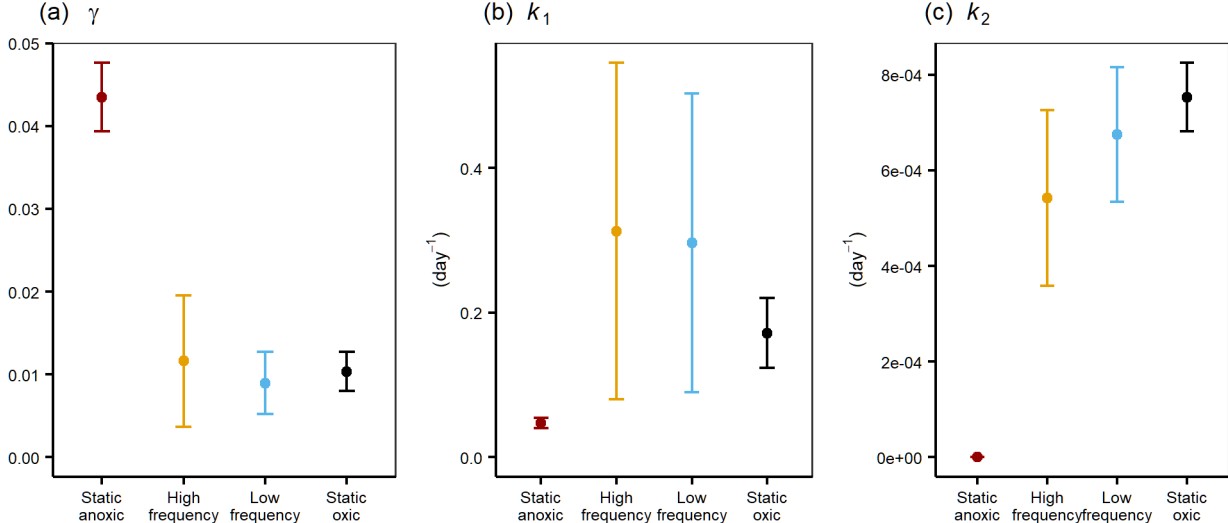

**Fig. 4. Effects of four redox treatments on coefficient estimates of a two-pool decomposition model applied to CO$_2$ fluxes from a**
**tropical redox incubation study. Modeled coefficients include: (a) $\gamma$, (b) $k_1$ and (c) $k_2$, where $\gamma$ is the coefficient that regulates initial**
**C partitioning into fast- and slow-cycling pools, and $k_1$ and $k_2$ are decomposition constants for the fast- and slow-cycling organic**
**carbon pools, respectively. Error bars indicate standard deviations of the coefficient distributions obtained through Bayesian**
**optimization.**





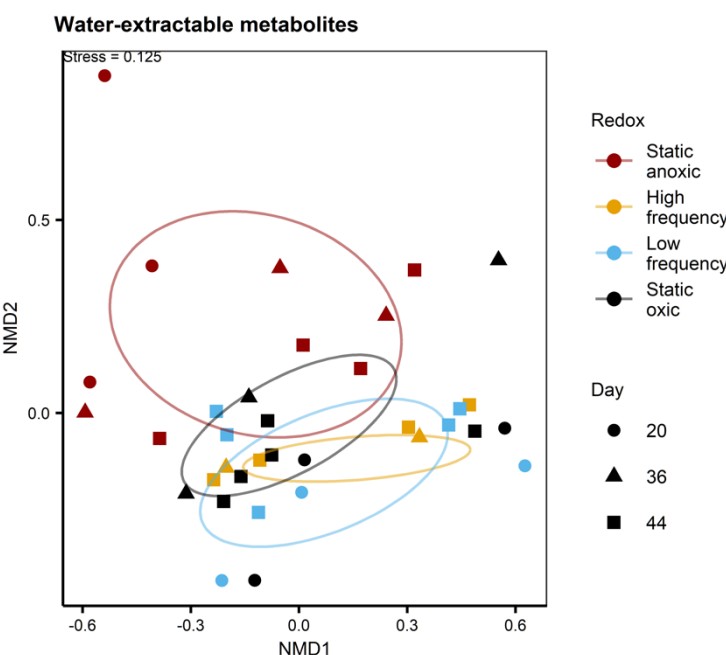

**Fig. 5. Non-metric multidimensional scaling (NMDS) plot comparing the composition of water-extractable organic matter among redox treatments and among sampling days. Data were derived from FTICR-MS analysis. The eclipse indicates the standard deviation of each redox treatment. High frequency treatment from day 20 was not included. Three outliers were removed. See Materials and Methods for details.**




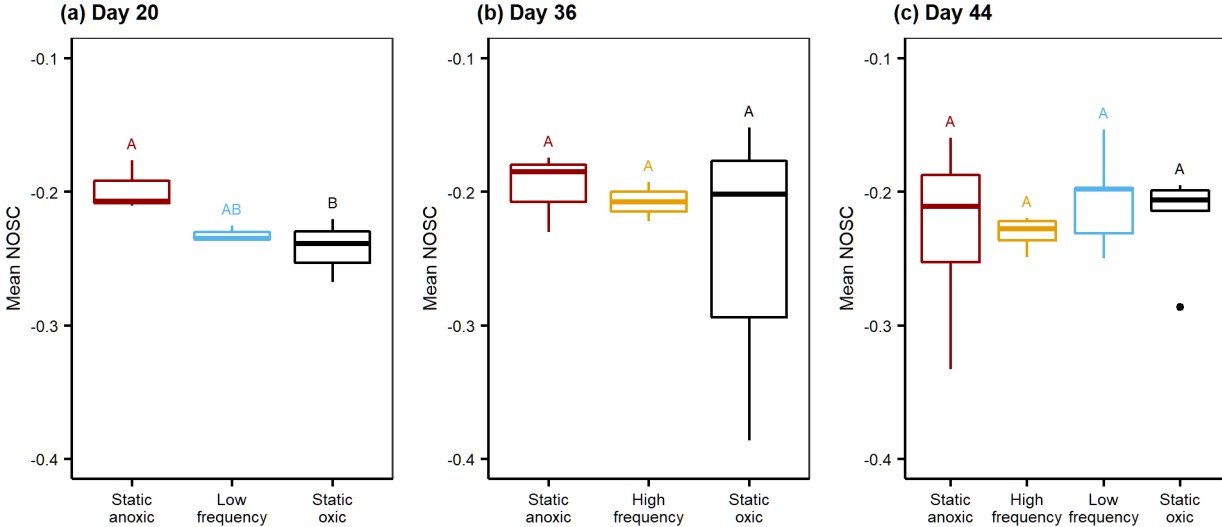

Fig. 6. The mean nominal oxidation state of carbon (NOSC) value of water-extractable organic matter from a tropical soil incubated
under four redox regimes collected on (a) day 20, (b) day 36, and (c) day 44 of the experiment. Data were derived from FTICR-MS
analysis. Boxplot whiskers represent 1.5 times the interquartile range of data. Different letters indicate significant difference at $\alpha$ =
0.05 level (Tukey's tests). On days 20 and 36, $n$ = 3 per treatment. On day 44, $n$ = 5 per treatment. High frequency treatment from
day 20 was not included. Three outliers were removed. See Materials and Methods for details.





**Fig. 7. Percentages of water-extractable organic matter that are unique to Day 20, unique to Day 44, and observed on both days in the static anoxic (top) and static oxic (bottom) treatment. Data were derived from FTICR-MS analysis. Results are grouped by compound classes. The number of compounds in each class is listed above the x axis.**