# Peer review of "Differential effects of redox conditions on the decomposition of litter and soil organic matter"

_Biogeosciences, 2020_

## Referee Comment (RC1) · Tida Ge (Referee) · 6 May 2020

This manuscript studied litter and native soil organic matter decomposition under different redox conditions. Their results showed decomposition of litter and native soil organic matter responded differently to redox manipulation. They revealed microbial consumption of organic matter was more oxidized compounds under anoxic condition. Their results also suggest recalcitrant organic C pool rather than labile organic C pool was significantly affected by redox condition. These findings are interesting and contribute to our knowledge of C dynamics under redox fluctuations. This manuscript is well-written and is recommended to publish after a minor revision. Only several suggestions here for improvement: 1. when describing the result from Fig. 6, please also describe and discuss the NOSC change over time within the same treatment. I

assume the NOSC were same across all treatments at the beginning on Day 0. On Day 44 NOSC were still same across all treatments. So NOSC in all treatments decreased from Day 0 to Day 44 with the same level. Just under static anoxic condition this decrease was slower than other conditions. You may discuss on this. 2. you may calculate priming effect of SOM decomposition caused by litter addition and compare between four redox conditions. 3. in Fig. 4 $\gamma$ was much higher in static anoxic condition than other conditions. Maybe add discussion on this.

―――――――――――――――――

---

## Referee Comment (RC2) · Anonymous Referee #3 · 7 Jul 2020

The present manuscript investigate the differential effects of redox conditions on the decomposition of litter and SOM over 44 days. The results indicate that substrate source—freshly added litter vs. native organic matter—plays an important role in the redox sensitivity of organic matter decomposition. I have two major comments: 1. Bhattacharyya et al. (2018) have reported the coupled cycling of iron and carbon using the same redox microcosm experiments. Results indicated that redox effected the specie of Fe and the Fe−OM interactions in tropical soils, thus effect the bioavailability of SOM. I would like to see some results and discussions about the the transformation of Fe during redox fluctuations and its role in the decomposition of litter and SOM. Especially, the chemical component of WEOM may be highly influenced by iron cycle during redox fluctuations.

[Figure]
Interactive
comment

2. I don't understand why the author use G-test to delete so many identified molecular formulas (over 60%).

3. It seems from Table S2 that the repeatability of the FT-ICR MS results is poor. Three of the 12 samples showed significant outliers. I think this is very likely due to the extraction method used in this study. It is very hard to guarantee the representation of the bulk soil and litter samples only using 100 mg soil in WEOM extraction. And it is impossible to control the DOC concentration in FT-ICR MS analysis for all samples. So it is cannot exclude the differences induced by DOC concentration in FT-ICR MS analysis.

In all, I don't think the manuscript can be published in its present form.

---

## Author Comment (AC1) · 13 Jul 2020

Referee #1 Comments:

This manuscript studied litter and native soil organic matter decomposition under different redox conditions. Their results showed decomposition of litter and native soil organic matter responded differently to redox manipulation. They revealed microbial consumption of organic matter was more oxidized compounds under anoxic condition. Their results also suggest recalcitrant organic C pool rather than labile organic C pool was significantly affected by redox condition. These findings are interesting and contribute to our knowledge of C dynamics under redox fluctuations. This manuscript is well-written and is recommended to publish after a minor revision.

**Thank you for the encouragement!**

Only several suggestions here for improvement:
1. when describing the result from Fig. 6, please also describe and discuss the NOSC change over time within the same treatment. I assume the NOSC were same across all treatments at the beginning on Day 0. On Day 44 NOSC were still same across all treatments. So NOSC in all treatments decreased from Day 0 to Day 44 with the same level. Just under static anoxic condition this decrease was slower than other conditions. You may discuss on this.

**We did not find significant changes of NOSC values between day 20 and 44. We have added this result to the text (Line 290). We agree with the reviewer that the mean NOSC value would start with the same value and likely decrease over time due to the consumption of relatively oxidized compounds. Unfortunately, we did not analyze samples from day 0 with the FTICR-MS method. Thus, we don't have the capacity to compare the NOSCs between day 0 and other days.**

2. you may calculate priming effect of SOM decomposition caused by litter addition and compare between four redox conditions.

**Thank you for this suggestion. We did not include microcosms with only soil. Therefore, it is impossible to estimate the priming effects based on our current data.**

3. in Fig. 4 γ was much higher in static anoxic condition than other conditions. Maybe add discussion on this.

**We will add the following text in the discussion on Line 360:**
**"Our modeling analysis also shows that the size of the fast-cycling pool was larger in the static anoxic treatment than in other treatments, which is consistent with the observation that a high litter decomposition rate was maintained for an extended period time under anoxic conditions. Larger fast-cycling pool was also in line with the increased DOC concentrations under anoxic conditions. These findings support our interpretation that Fe reduction released labile C from organo-mineral complexes, which could be degraded anaerobically."**

---

## Author Comment (AC2) · 13 Jul 2020

Referee #3 Comments:

The present manuscript investigate the differential effects of redox conditions on the decomposition of litter and SOM over 44 days. The results indicate that substrate sourceă˘AˇT freshly added litter vs. native organic matter â˘AˇT plays an important role in the redox sensitivity of organic matter decomposition. I have two major comments: 1. Bhattacharyya et al. (2018) have reported the coupled cycling of iron and carbon using the same redox microcosm experiments. Results indicated that redox effected the specie of Fe and the Fe−OM interactions in tropical soils, thus effect the bioavailability of SOM. I would like to see some results and discussions about the the transformation of Fe during redox fluctuations and its role in the decomposition of litter and SOM. Especially, the chemical component of WEOM may be highly influenced by iron cycle during redox fluctuations.

> *We agree with the reviewer that Fe transformation plays an important role in organic matter dynamics. We are preparing a separate manuscript focusing on Fe speciation using XANES and NanoSIMS methods. Although these results are beyond the scope of this study, we will discuss the potential roles of Fe reduction in organic matter decomposition. We will add a new paragraph on line 355 with the following text:*
> *"Iron reduction likely played a key role in sustaining high decomposition rates under low redox conditions (Huang et al. 2020; Chen et al. 2020). High abundance and activity of Fe-reducing bacteria have been commonly observed in soils from the same ecosystem (Teh et al. 2008; DeAngelis et al. 2010; Dubinsky et al. 2010). Using unlabeled litter additions and similar redox regimes, a companion study (Bhattacharyya et al. 2018) observed significant Fe reduction under static anoxic conditions. Thus we presume that iron reduction was responsible for the higher DOC levels in the static anoxic treatments, as it can trigger the dissolution of organo-Fe complexes and mobilization of organic matter (Pan et al. 2016; Huang et al. 2020). Iron reduction is also thermodynamically favorable when coupled with the decomposition of relatively oxidized compounds (Keiluweit et al. 2016)."*
>
> *In addition, the following text will be added after Line 362:*
> *"Larger fast-cycling pool was also in line with the increased DOC concentrations under anoxic conditions. These findings support our interpretation that Fe reduction released labile C from organo-mineral complexes, which was then degraded anaerobically."*
>
> *We also want to clarify that this study was conducted using $^{13}$C-enriched litter, while Bhattacharyya et al. (2018) used unlabeled litter (Lines 111-123). Thus, this study presents a new data set.*

2. I don't understand why the author use G-test to delete so many identified molecular formulas (over 60%).

> *The FTICR-MS data have relatively high variability in both the intensities and presence/absence of compounds. Absence of compounds can be caused by random errors or true biological effects, making interpretation difficult. We adopted a conservative method in using a G-test to limit the chance of including random errors (Webb-Robertson et al., 2010; Bramer and White, 2019). Specifically, we only considered peaks that were present in at least 2 replicates of each combination of treatment and sampling day. This method is well accepted for analyzing FTICR-MS and other high resolution MS data (Varnum et al. 2012; Nakayasu*

*et al. 2016; Piehowski et al. 2020). We note that the G-test was only used for the analysis presented in Figure 7, which focused on the presence/absence patterns. Other analyses such as the NMDS and NOSC comparison were not subject to the G-test.*

3. It seems from Table S2 that the repeatability of the FT-ICR MS results is poor. Three of the 12 samples showed significant outliers. I think this is very likely due to the extraction method used in this study. It is very hard to guarantee the representation of the bulk soil and litter samples only using 100 mg soil in WEOM extraction. And it is impossible to control the DOC concentration in FT-ICR MS analysis for all samples. So it is cannot exclude the differences induced by DOC concentration in FT-ICR MS analysis.

*We are confident about our analytical procedures and somewhat puzzled by the comment about Table S2. This may be because the table legend did not specify that the values presented were the sum of multiple replicates--and that multiple samples were run for each sampling day and treatment. A total of 44 samples were analyzed with FTICR-MS. Four were obvious outliers with less than 200 peaks (Lines 158-161). Only one outlier was identified using the robust Mahalanobis distance approach (Lines 161-166).*

*We will modify the table to show the means and standard errors of individual replicates. The new table and table legend will be:*
*"Mean and standard error of water-extractable molecular formulas identified by FTICR-MS analysis per sampling day and treatments. For each treatment and harvest timepoint, replicate samples were analyzed (3, 3, and 5 replicates on days 20, 36, and 44, respectively)"*

| Sampling Day | Static anoxic | High frequency | Low frequency | Static oxic |
|---|---|---|---|---|
| 20 | 1883±211 | N/A[a] | 1883±509 | 1742±490 |
| 36 | 1513±446 | 1826±653 | N/A[b] | 1769±399 |
| 44 | 1315±195 | 1751±302 | 1805±280 | 1782±269 |

*a Treatment was removed due to presence of outliers. See Methods section for details.*
*b Treatment was not included in the original experimental design.*

*The PNNL-EMSL DOE user facility runs 1000s of FTICR-MS analyses per year, and for soil samples like ours, typically uses 100 mg of substrate for extracting organic matter prior to FTICR-MS analysis. The facility achieves great consistency and repeatability as demonstrated by their peer-reviewed publications (e.g., Tfaily et al. 2015; Tfaily et al. 2017; Boye et al. 2017).*

*Regarding extraction efficiency/consistency, we apologize for not mentioning in the methods that the ion accumulation time on the FTICR-MS instrument was adjusted for all samples to keep similar numbers of ions in the ICR cell. This is a standard practice at the EMSL facility and helps to control for differences in DOC among samples (Boye et al. 2017; Graham et al. 2017). The following text will be added on Line 151:*

*"Ion accumulation time was optimized for all samples to account for differences in DOC concentration (Graham et al. 2017; Boye et al. 2017)."*

*Differences in DOC concentration were unlikely to drive the observed patterns in DOC chemistry (Lines 308-310). The DOC composition showed strong temporal trends: the static anoxic treatment was different from other treatments on day 20 (Figs. 5, 6, 7, and S4), while these differences disappeared on day 44. In contrast, DOC concentrations were higher in the static anoxic treatment than in other treatments on both days 20 and 44 (Fig. S5). These results demonstrate that the observed differences in DOC composition are robust and not driven by variation in DOC concentration among treatments.*

In all, I don't think the manuscript can be published in its present form.

*We hope that we have addressed your concerns.*

References

Bhattacharyya, A., Campbell, A. N., Tfaily, M. M., Lin, Y., Kukkadapu, R. K., Silver, W., Nico, P. S., and Pett-Ridge, J.: Redox fluctuations control the coupled cycling of iron and carbon in tropical forest soils, Environmental Science & Technology, 10.1021/acs.est.8b03408, 2018.

Bramer, L., and White, A.: ftmsRanalysis: Analysis and visualization tools for FT-MS data. R package version 1.0.0. https://github.com/EMSL-Computing/ftmsRanalysis, 2019.

Chen, Chunmei, Steven J. Hall, Elizabeth Coward, and Aaron Thompson. Iron-mediated organic matter decomposition in humid soils can counteract protection. Nature communications 11, 2255, 2020.

DeAngelis, K. M., Silver, W. L., Thompson, A. W., and Firestone, M. K.: Microbial communities acclimate to recurring changes in soil redox potential status, Environmental Microbiology, 12, 3137-3149, 10.1111/j.1462-2920.2010.02286.x, 2010.

Dubinsky, E. A., Silver, W. L., and Firestone, M. K.: Tropical forest soil microbial communities couple iron and carbon biogeochemistry, Ecology, 91, 2604-2612, 10.1890/09-1365.1, 2010.

Ginn, B., Meile, C., Wilmoth, J., Tang, Y., and Thompson, A.: Rapid Iron Reduction Rates Are Stimulated by High-Amplitude Redox Fluctuations in a Tropical Forest Soil, Environmental Science & Technology, 51, 3250-3259, 10.1021/acs.est.6b05709, 2017.

Huang, W., Ye, C., Hockaday, W. C., & Hall, S. J. (2020). Trade-offs in soil carbon protection mechanisms under aerobic and anaerobic conditions. Global Change Biology, 26, 3726-3737.

Keiluweit, M., Nico, P. S., Kleber, M., and Fendorf, S.: Are oxygen limitations under recognized regulators of organic carbon turnover in upland soils?, Biogeochemistry, 10.1007/s10533-015-0180-6, 2016.

Nakayasu, E. S., Nicora, C. D., Sims, A. C., Burnum-Johnson, K. E., Kim, Y. M., Kyle, J. E., ... & Jacobs, J. M. (2016). MPLEx: a robust and universal protocol for single-sample integrative proteomic, metabolomic, and lipidomic analyses. MSystems, 1(3).

Pan, W., Kan, J., Inamdar, S., Chen, C., and Sparks, D.: Dissimilatory microbial iron reduction release DOC (dissolved organic carbon) from carbon-ferrihydrite association, Soil Biology and Biochemistry, 103, 232-240, 10.1016/j.soilbio.2016.08.026, 2016.

Piehowski, P. D., Zhu, Y., Bramer, L. M., Stratton, K. G., Zhao, R., Orton, D. J., ... & Webb-Robertson, B. J. M. (2020). Automated mass spectrometry imaging of over 2000 proteins from tissue sections at 100-μm spatial resolution. Nature communications, 11(1), 1-12.

Teh, Y. A., Dubinsky, E. A., Silver, W. L., and Carlson, C. M.: Suppression of methanogenesis by dissimilatory Fe(III)-reducing bacteria in tropical rain forest soils: Implications for ecosystem methane flux, Global Change Biology, 14, 413-422, 10.1111/j.1365-2486.2007.01487.x, 2008.

Tfaily, M., Chu, R., Tolić, N., Roscioli, K., Anderton, C., Paša-Tolić, L., Robinson, E., Hess, N. (2015). Advanced Solvent Based Methods for Molecular Characterization of Soil Organic Matter by High-Resolution Mass Spectrometry Analytical Chemistry 87(10), 5206-5215. https://dx.doi.org/10.1021/acs.analchem.5b00116

Tfaily, M., Chu, R., Toyoda, J., Tolić, N., Robinson, E., Paša-Tolić, L., Hess, N. (2017). Sequential extraction protocol for organic matter from soils and sediments using high resolution mass spectrometry Analytica Chimica Acta 972(Agric. Ecosyst. Environ. 104 2004), 54-61. https://dx.doi.org/10.1016/j.aca.2017.03.031

Varnum, S. M., Webb-Robertson, B. J. M., Pounds, J. G., Moore, R. J., Smith, R. D., Frevert, C. W., ... & Wunschel, D. (2012). Proteomic analysis of bronchoalveolar lavage fluid proteins from mice infected with Francisella tularensis ssp. novicida. Journal of proteome research, 11(7), 3690-3703.

Webb-Robertson, B.-J. M., McCue, L. A., Waters, K. M., Matzke, M. M., Jacobs, J. M., Metz, T. O., Varnum, S. M., and Pounds, J. G.: Combined Statistical Analyses of Peptide Intensities and Peptide Occurrences Improves Identification of Significant Peptides from MS-Based Proteomics Data, Journal of Proteome Research, 9, 5748-5756, 10.1021/pr1005247, 2010.